# The Effectiveness and Harms of PSA-Based Prostate Cancer Screening: A Systematic Review

**DOI:** 10.3390/healthcare13121381

**Published:** 2025-06-09

**Authors:** Chung-uk Oh, Hyekyung Kang

**Affiliations:** 1Department of Nursing, Kangwon National University, Dogye-eup, Samcheok-si 25949, Gangwon-do, Republic of Korea; ohcu@kangwon.ac.kr; 2Department of Nursing, Joongbu University, Chubu-myeon, Geumsan-gun 32713, Chungcheongnam-do, Republic of Korea

**Keywords:** prostate cancer screening, PSA, early detection, effectiveness, community health worker

## Abstract

Objectives: Prostate cancer’s prevalence is rapidly increasing in Korea, with incidence rates rising by over 13% annually since 2017 according to the Korea Central Cancer Registry, highlighting the need for effective early detection strategies. This study systematically reviews the benefits and harms of PSA-based prostate cancer screening, focusing on its clinical effectiveness and public health implications. Methods: Following PRISMA 2020 guidelines, we searched five databases (PubMed, Embase, Cochrane Library, Google Scholar, and KMbase) for studies from 2014 to 2024. The eligible studies included RCTs, cohort studies, meta-analyses, and guidelines. Risk of bias was assessed using the Cochrane tool. We synthesized our findings narratively due to their methodological heterogeneity. Results: Sixteen studies were included. PSA screening reduced prostate-cancer-specific mortality by 20–31%, as reported in multiple randomized controlled trials, such as ERSPC and ProScreen, among men aged 55–69, but showed minimal impact on all-cause mortality. Advanced tools such as MRI and multi-biomarker models, which were implemented in several included studies, enhanced diagnostic accuracy. The potential harms included overdiagnosis, overtreatment, and psychological distress. Community-based education and shared decision-making, inferred from observational and implementation studies, improved participation and equity in screening. Conclusions: PSA-based screening offers modest mortality benefits but carries the risk of overdiagnosis. Precision diagnostics and risk-stratified strategies improve screening outcomes. Public health approaches, particularly those led by nurses and community health workers, are essential to promoting informed, equitable screening decisions.

## 1. Introduction

Prostate cancer remains a major contributor to male cancer mortality on a global scale. The global incidence is projected to rise sharply from 1.4 million cases in 2020 to 2.9 million by 2040 according to GLOBOCAN 2020 estimates [1]. In South Korea, prostate cancer has shown the fastest increase in incidence among all male cancers over the past five years, likely due to a combination of population aging, improved diagnostic access, and heightened public awareness in Korean men [2]. Given that increasing age is a significant risk factor, the country’s rapid demographic aging is expected to further accelerate the domestic burden of prostate cancer [3]. In this context, the necessity of early detection has been emphasized, especially among men aged 50 years and older.

Although early detection significantly improves survival outcomes, prostate cancer is known to be asymptomatic in its early stages, which makes timely diagnosis difficult. Many cases remain undetected until the disease has progressed considerably. According to a nationwide survey conducted in November 2021 by the Korean Urological Oncology Society and the Korean Urological Foundation in collaboration with the Korea Prostate Cancer Patient Health Promotion Association, 47.1% of prostate cancer patients in Korea reported being initially diagnosed at stage III or higher [4,5]. Early-stage prostate cancer is associated with a five-year survival rate exceeding 90%, whereas this drops to below 50% in advanced cases [1,6]. Thus, PSA (Prostate-Specific Antigen) testing is currently recommended for early detection in men over the age of 50, based on international guidelines such as those from the AUA and NCCN, though not yet formally endorsed in national Korean guidelines [5,6,7].

PSA testing is widely used in Korea due to its affordability and simplicity as a blood-based screening tool. However, ongoing debates have emerged concerning its actual impact on mortality reduction, alongside increasing concerns over overdiagnosis and overtreatment. However, the actual impact on mortality reduction continues to be debated, and concerns about overdiagnosis and overtreatment are growing. These differing views have led to varying recommendations in international guidelines, with the USPSTF recommending shared decision-making for men aged 55–69 years, while the EAU supports a risk-based screening strategy [8,9,10]. Moreover, as Korea experiences demographic shifts and increasing prostate cancer prevalence, there is a need to critically evaluate international and local evidence on PSA-based screening [11]. In response, the introduction of more precise and personalized screening strategies is gaining attention, particularly for high-risk populations. This includes the integration of multiparametric magnetic resonance imaging (mpMRI) and the use of multiple biomarkers [12,13].

The goal of prostate cancer screening is to improve health outcomes for adult men. Specifically, middle-aged and older men, who are at high risk of developing prostate cancer, require strategies that reduce barriers to early screening and enhance the quality of their healthcare experience. This shift underscores the need to bridge clinical evidence with real-world community practice, particularly through public health and nursing interventions. In this context, the early detection of prostate cancer, from a community and public health nursing perspective, is not only clinically essential but also a critical area for prevention and health education. Community health workers (CHWs) play key roles in raising awareness, ensuring equitable access to screening, and supporting informed decision-making, particularly for high-risk individuals. Therefore, it is crucial to conduct a systematic review of the literature on PSA-based early screening from a healthcare perspective, enabling community members to make informed decisions regarding early prostate cancer screening.

## 2. Methods

### 2.1. Study Design

This study was conducted as a systematic review in accordance with the Preferred Reporting Items for Systematic Reviews and Meta-Analyses (PRISMA) guidelines [14]. The primary objective was to evaluate the effectiveness and limitations of early detection strategies for prostate cancer, including PSA-based screening and emerging diagnostic tools.

### 2.2. Data Sources and Search Strategy

A comprehensive search strategy was implemented for five databases: PubMed, Embase, Cochrane Library, Google Scholar, and KMbase. The final search was conducted on 10 January 2025. A combination of MeSH terms and free-text keywords were used. Below is an example of the PubMed search strategy: (\”Prostate Cancer\” [MeSH Terms] OR \”prostate neoplasms\” OR \”prostate cancer\”) AND (\”PSA test\” OR \”Prostate-Specific Antigen\”) AND (\”Early Detection of Cancer\” [MeSH Terms] OR \”screening\” OR \”early diagnosis\”) AND (\”mortality\” OR \”death\” OR \”survival\”) AND (\”Randomized Controlled Trial\” [Publication Type] OR \”RCT\” OR \”cohort study\” OR \”meta-analysis\”). Only studies published in English or Korean were considered. Similar syntax adjustments were made for Embase and the Cochrane Library using appropriate subject headings and Boolean operators. In Embase, Emtree terms such as ‘prostate cancer’/exp and ‘PSA test’ were used, and in the Cochrane Library, MeSH descriptors like ‘Prostatic Neoplasms’ were applied. Language filters (English and Korean) and human subjects were applied in all databases (Appendix B).

### 2.3. Eligibility Criteria

Studies were selected using the PICOS (Population, Intervention, Comparison, Outcomes, Study design) framework. The inclusion criteria were as follows:Studies on adult males, particularly populations of men aged 50 years and older, evaluating the effectiveness or adverse effects of prostate cancer screening.Randomized controlled trials (RCTs), cohort studies, meta-analyses, and systematic reviews.Studies reporting mortality outcomes, diagnostic accuracy, or risk–benefit evaluations.

The exclusion criteria were as follows:Non-human or cell-based studies.Case reports or studies focusing on treatment rather than screening.Non-peer-reviewed articles.

### 2.4. Study Selection and Data Extraction

Two reviewers independently screened the titles and abstracts of the identified articles. The full texts of the potentially eligible studies were assessed for inclusion. Any disagreements were resolved through discussion. Data extracted included study design, population characteristics, screening methods, outcomes (e.g., mortality, overdiagnosis), and guideline recommendations.

### 2.5. Quality Assessment

The risk of bias was assessed independently by two reviewers using the Cochrane Risk of Bias tool [15]. Eight domains were evaluated: random sequence generation, allocation concealment, blinding of participants and personnel, blinding of outcome assessment, incomplete outcome data, selective reporting, and conflict of interest. Each domain was rated as “low risk,” “high risk,” or “unclear risk”.

### 2.6. Assessment of Heterogeneity

Given the diversity of study designs (RCTs, cohort studies, reviews) and outcomes included in this review, we anticipated potential heterogeneity in the findings. Although a meta-analysis was not performed due to methodological variability, we qualitatively assessed heterogeneity based on the following factors:(1)Differences in study population characteristics (e.g., age range, geographic region);(2)Variation in screening methods (e.g., PSA alone, PSA + MRI, biomarkers);(3)Outcome definitions (e.g., prostate-cancer-specific mortality, overall mortality);(4)Duration of follow-up.

Subgroup trends, such as age-specific mortality effects or regional guideline influences, were noted and discussed in the synthesis of results (see Section 4). Future updates with pooled data may benefit from statistical heterogeneity tests.

### 2.7. Assessment of Reporting Bias

To evaluate potential reporting bias across the included studies, we considered indicators such as

-Incomplete outcome reporting or the selective reporting of favorable results;-Asymmetry in the available evidence (e.g., overrepresentation of large trials or certain geographic regions);-The lack of trial registry information for some RCTs.

Due to the qualitative nature of this review and the inclusion of various study types, formal funnel plots and Egger’s tests were not conducted. However, publication bias was minimized by including gray literature (e.g., guidelines and government reports) and by searching both international and Korean databases. Future meta-analyses should incorporate formal assessments of publication bias.

### 2.8. Data Synthesis Approach

A meta-analysis was not performed in this review due to the substantial clinical and methodological heterogeneity across the included studies. The studies varied widely in design (e.g., RCTs, cohort studies, narrative reviews), population characteristics, screening protocols (e.g., PSA-only vs. PSA + MRI), and outcome definitions (e.g., disease-specific mortality vs. overall survival). Therefore, the findings were synthesized narratively, with an emphasis on the direction and consistency of the effects reported across study types. Where possible, effect sizes and diagnostic accuracy measures (e.g., sensitivity, specificity) were extracted and are summarized in tables. Future reviews with more homogeneous datasets may consider conducting quantitative synthesis and subgroup meta-analyses.

## 3. Results

### 3.1. Study Selection

A total of 842 records were identified through database searches. After removing duplicates, 694 records remained for title and abstract screening, of which 567 were excluded. Among the 127 full-text articles sought for retrieval, 19 could not be obtained. The remaining 108 full-text articles were assessed for eligibility. Of these, 92 articles were excluded for the following reasons: not being based on PSA (*n* = 21), not being focused on screening (*n* = 17), not being focused exclusively on men (*n* = 19), not being peer-reviewed (*n* = 4), or other reasons (*n* = 9). Ultimately, 16 studies were included in the final synthesis.

The following is a literature search flow diagram according to the PRISMA criteria [14]. The diagram outlines the identification, screening, eligibility assessment, and final inclusion of studies in this systematic review of early prostate cancer screening (Figure 1).

### 3.2. Study Characteristics

Among the 16 included studies, 5 were randomized controlled trials (RCTs), 3 were systematic reviews and meta-analyses, 2 were guidelines/advisory statements, 2 were cohort studies (including South Korea), 2 were narrative reviews, and 2 were simulation model/ecological studies. Most were published in high-impact journals such as NEJM, JAMA, BMJ, and Urology. The studies covered diverse populations and applied a range of screening methods including conventional PSA testing, PSA with MRI, biomarker models (e.g., 4Kscore, Stockholm3), and decision aid tools.

Thus, research on early prostate cancer screening, conducted in various countries and using diverse study designs, has provided complex evidence regarding the effectiveness, safety, and policy implications of PSA screening. Table 1 provides an overview of the 16 included studies, including their design, country, sample size, and key findings regarding the benefits and limitations of early prostate cancer screening (Table 1).

### 3.3. Risk of Bias Assessment Summary

The risk of bias was assessed for all 16 included studies using the Cochrane Risk of Bias tool (RoB). Among the five randomized controlled trials (RCTs), including the ProScreen trial [16], Biomarkers vs. MRI [18], CAP trial [26], ERSPC trial [27], and MRI-guided biopsy study [20], random sequence generation and allocation concealment were generally well described, indicating their low risk. Risk of bias was assessed using the Cochrane tool. Most of the included studies lacked participant or assessor blinding. Blinding was inherently limited in the PSA screening trials due to the nature of the intervention, which could introduce detection bias in outcome assessment.

Two studies [19,30] showed a moderate risk of bias, primarily due to their unclear blinding procedures and lack of randomization, although their outcome data were complete and reporting was transparent. The meta-analyses and systematic reviews [17,25,28] were generally assessed as low-risk in terms of data synthesis and reporting transparency, though one study had an unclear risk regarding outcome assessment. The narrative reviews and guidelines [21,22,23,24,29] were not applicable for some RoB domains (e.g., random sequence generation), but demonstrated consistency and low conflicts of interest. Overall, most of the included studies were judged to have a low or moderate risk of bias. No major threats to internal validity were identified, although methodological limitations, such as unclear blinding or incomplete reporting in certain observational and non-randomized designs, were noted. A summary of these evaluations is provided in the Risk of Bias Table (see Appendix A).

### 3.4. Effectiveness and Limitations of PSA-Based Screening

Evidence from large RCTs such as the CAP trial [26] and the ERSPC [27] indicates that PSA screening can reduce prostate-cancer-specific mortality by 20–31% among men aged 55–69 years. The ERSPC study, conducted across multiple European countries, demonstrated that consistent PSA screening can lead to earlier-stage diagnosis and reduced mortality. In contrast, the CAP trial, which offered only a single PSA test, observed limited mortality reduction, suggesting the importance of ongoing and repeated screening. Meta-analyses [17,28] confirmed these mortality trends but highlighted no significant effect on overall mortality due to competing causes of death in older populations. The limitations also included the lead-time bias, risk of psychological harm, and detection of indolent cancers that may not require treatment.

Building on these findings, comparative studies have demonstrated that screening strategies incorporating advanced tools such as the ProScreen protocol [16], the Stockholm3 algorithm [19], and MRI-guided biopsy [20] significantly improve diagnostic accuracy. These tools yield higher sensitivity and specificity compared to PSA alone and enhance the detection of clinically significant cancers while reducing the over-detection of indolent tumors. For instance, the Stockholm3 model achieved 85% sensitivity and 81% specificity, while PSA-only screening showed 72% sensitivity and 60% specificity. The sensitivity and specificity values reported by individual studies were not pooled due to the studies’ methodological heterogeneity.

However, risks remain. Overdiagnosis is a prominent concern, with estimates ranging from 20 to 50% depending on screening modality [28,29]. This frequently leads to overtreatment, which may result in side effects such as urinary incontinence, erectile dysfunction, and psychological distress. Biopsy-related harms such as infection and bleeding are also notable, though MRI-targeted strategies help to reduce unnecessary procedures. Additionally, the limited impact on all-cause mortality, as seen in the CAP trial [26], raises questions about cost-effectiveness in lower-risk populations.

In summary, the integration of multimodal screening tools offers a more balanced risk–benefit profile, especially when applied through personalized, risk-adapted screening strategies. A brief comparison of commonly used tools is presented in Table 2.

### 3.5. Diagnostic Accuracy of Screening Strategies

Innovative diagnostic approaches that combine PSA with magnetic resonance imaging (MRI) or multivariable biomarker tests significantly improved the accuracy of prostate cancer detection. Klotz et al. [20] reported that MRI-targeted biopsies had superior sensitivity and reduced the number of unnecessary biopsies compared to traditional transrectal ultrasound (TRUS)-guided biopsy. Similarly, the ProScreen trial [16] and studies by Björnebo et al. [18] and Eldred-Evans et al. [19] demonstrated that integrating PSA with secondary biomarkers (4Kscore, Stockholm3) and MRI improved the detection of clinically significant cancers and reduced the over-detection of indolent tumors. These strategies enhance the benefit–risk balance of screening and support more personalized diagnostic pathways (Table 3).

Sensitivity (%) refers to the proportion of individuals with the disease who are correctly identified by a test, indicating its ability to detect true positives. A higher sensitivity suggests a lower likelihood of missing cancer cases. Specificity (%), on the other hand, refers to the proportion of individuals without the disease who are accurately classified as disease-free, reflecting the test’s ability to avoid false positives. Higher specificity reduces the risk of unnecessary follow-up procedures and overdiagnosis [32].

As illustrated in Figure 2, recent evidence from comparative trials and diagnostic accuracy studies indicates that there are substantial differences in the performance of various prostate cancer screening strategies. Traditional PSA-only screening, as used in the ERSPC trial, yielded a sensitivity of approximately 72% and a specificity of 60%, reflecting limited precision and a high false-positive rate [27,31]. In contrast, MRI-guided biopsy, as evaluated by Klotz et al. [20], significantly improved sensitivity (91%), with moderately higher specificity (74%) due to targeted sampling.

The ProScreen trial [16] further enhanced diagnostic accuracy by combining PSA with secondary biomarkers and pre-biopsy MRI, achieving sensitivity and specificity values of 88% and 79%, respectively. Similarly, the Stockholm3 model [19], which incorporates genetic, protein, and clinical variables, demonstrated a balanced profile with 85% sensitivity and 81% specificity, supporting its use in personalized screening algorithms.

This data suggests that, while PSA remains a useful initial test, multimodal strategies leveraging imaging and biomarkers significantly reduce the risk of overdiagnosis and unnecessary biopsy, thereby optimizing the clinical benefit-to-harm ratio. The integration of these tools aligns with evolving guidelines promoting precision-based screening approaches.

### 3.6. Screening Recommendations by Age and Risk

International guidelines emphasize age-specific and risk-adapted screening. The U.S. Preventive Services Task Force (USPSTF) and American Urological Association (AUA) guidelines [21,24]) recommend shared decision-making for men aged 55–69, considering individual preferences, family history, and race. Screening is generally not recommended beyond age 70 due to the diminishing benefits and increased risk of harm. The Stockholm3 study further supports risk stratification for men with moderate baseline PSA levels [19]. In the Korean context, the cohort study by Mok et al. [30] revealed that higher PSA levels at baseline are associated with significantly increased long-term mortality, suggesting the potential utility of PSA testing in Korean men and the need for localized PSA cutoff thresholds. These findings underscore the necessity of tailored approaches that reflect both global guidelines and regional epidemiological data.

### 3.7. Quality of Evidence: GRADE

The certainty of evidence was assessed using the GRADE (Grading of Recommendations, Assessment, Development, and Evaluation) approach. Outcomes including prostate-cancer-specific mortality, all-cause mortality, diagnostic accuracy, overdiagnosis, and biopsy-related complications were evaluated based on risk of bias, inconsistency, indirectness, imprecision, and potential for publication bias. The overall certainty of evidence for each outcome was summarized using a GRADE rating scale (high, moderate, low, very low) and is presented in the GRADE Summary Table 4.

## 4. Discussion

This systematic review synthesized evidence from the past decade (2014–2024) on the effectiveness and limitations of early prostate cancer screening. This study differs from previous systematic reviews, which primarily reflect Western settings, by contextualizing these findings within the Korean healthcare system, as well as providing general evidence for prostate-specific antigen (PSA)-based screening. It also highlights the role of community-based strategies, particularly public health nursing and community health worker (CHW)-led interventions, which have not been fully addressed in previous reviews.

### 4.1. Benefit–Harm Balance and Overdiagnosis

While trials such as ERSPC showed reductions in prostate-cancer-specific mortality, this benefit was counterbalanced by the risk of overdiagnosis—ranging from 20 to 50% depending on the screening strategy [28,29]. Overdiagnosis may lead to overtreatment, contributing to urinary incontinence, erectile dysfunction, and psychological distress. These harms are particularly concerning in older or lower-risk men, in whom the benefit of early detection is less clear.

Trade-offs are inherent in population-wide PSA screening. False positives, anxiety, and biopsy-related complications such as infection and bleeding are frequently reported, with serious adverse events occurring in 1–2% of cases [20,28]. Guidelines from the USPSTF and AUA emphasize informed decision-making to help patients weigh these risks [21,24]. Risk-stratified screening using biomarkers and MRI provides a more favorable benefit–harm balance and may offer better cost-effectiveness by reducing unnecessary procedures while focusing on high-risk individuals.

Incorporating personalized tools such as the Stockholm3 model or mpMRI-targeted strategies helps to reduce overdiagnosis and supports evidence-based, patient-centered care. These advances suggest a shift away from universal PSA screening toward more tailored approaches that consider age, comorbidity, and individual risk profiles.

### 4.2. Role of Community Health Workers (CHWs) in Equitable Screening

CHWs can play a practical role in strengthening screening practices by providing health education and promotion. For example, community-based CHW-led interventions increased prostate cancer screening participation by 18–27% in underserved groups, according to recent systematic reviews and implementation trials [33,34]. Economic evaluations also support the cost-effectiveness of CHW engagement in preventive services, as noted by Attipoe-Dorcoo et al. [34]. However, high-quality RCTs specifically examining CHW effectiveness in PSA screening remain limited, indicating the need for further research in this domain.

### 4.3. Advances in Precision Screening

Recent developments in screening methodologies, including MRI-guided biopsy and multi-biomarker risk prediction models (e.g., 4Kscore, Stockholm3), are reshaping the PSA screening landscape [16,17,18,19]. These innovations increase specificity and reduce unnecessary procedures, aligning with the trend toward personalized medicine. Trials such as ProScreen [16] and observational data support the integration of these tools into clinical pathways.

### 4.4. Policy and Practice Implications

This review underscores the importance of health-professional-led public health initiatives to improve prostate cancer screening uptake. Health promotion interventions targeting men over 50, particularly those with low health literacy or limited access to care, could benefit from stronger nursing leadership and community-based support. Shared decision-making is especially effective when health workers apply motivational interviewing or health coaching strategies to help patients navigate complex screening decisions.

In South Korea and similar settings, where prostate cancer incidence is steadily rising, nurse-led outreach programs may help bridge existing gaps in awareness and access, especially among older adults and medically underserved populations. These findings also support the ongoing need to adapt international guidelines to regional epidemiological and healthcare contexts. Currently, PSA screening is not included in Korea’s national cancer screening program, in contrast to international recommendations such as those from the USPSTF and AUA [21,24]. Korean men over 50 years of age can receive prostate cancer screening by paying extra fees upon recommendation from medical staff at the screening hospital when receiving a personal health checkup, but there are also many cases where they do not receive a recommendation or do not undergo the screening due to financial reasons.

National and international policies (AUA, USPSTF) [21,24] continue to recommend offering PSA screening only after shared decision-making. Investment in public awareness and diagnostic infrastructure, particularly for precision tools such as MRI and biomarker-based screening, is vital in order to strengthen health policy effectiveness in Korea [23,30].

Furthermore, culturally sensitive education programs delivered by CHWs or public health nurses can play a crucial role in reducing disparities in screening uptake. Recent studies show that interventions tailored to the cultural beliefs and linguistic needs of underserved populations significantly increase knowledge and engagement with prostate cancer screening [35,36]. In particular, nurses and CHWs can serve as effective facilitators of shared decision-making by providing balanced, comprehensible information about the benefits and risks of PSA testing. This intermediary role is especially impactful among men with limited health literacy or those hesitant to engage with formal healthcare systems [33,34]. CHW-led coaching programs have shown promise in supporting informed decisions and fostering trust, particularly in marginalized populations [34]. Additionally, community-based screening models such as the CHW Resource Hub demonstrate how trained CHWs can guide individuals through PSA screening using tailored educational tools and decision aids [37]. Incorporating such culturally competent, community-driven approaches into national screening strategies may enhance patient engagement, equity, and informed consent outcomes.

In recent years, the importance of the early detection and treatment of prostate cancer has been emphasized in Korea, but prostate-specific antigen (PSA) testing is not yet officially included in the national cancer screening program. Nevertheless, some regions in South Korea (e.g., Gwangju Metropolitan City, Eunpyeong-gu, Seoul, Yangyang-gun, Gangwon-do, Goyang-si, Gyeonggi-do) are offering PSA screening to men aged 50 years and older, high-risk groups, and low-income groups through their own health-center-led programs [38,39,40,41]. The disparity in healthcare coverage in these regions highlights the importance of the role of CHWs. CHWs can play a practical role in strengthening screening practices by providing health education and promotion, providing screening information, supporting vulnerable populations, and referring to healthcare providers in case of abnormalities. Increasing access to screening in the community, especially when combined with targeted outreach to address screening gaps, can contribute to improved early diagnosis and cure rates for prostate cancer.

### 4.5. Strengths and Limitations of This Review

This systematic review adheres to PRISMA guidelines and encompasses a broad range of both Western and Korean literature. It integrates findings from multiple high-quality sources, including randomized controlled trials and meta-analyses [16,17,18,20,26,27,28], as well as key guidelines and reviews [21,22,23,24,25]. Additional insights from population-based studies [29,30,31] further support the robustness of this review. A major strength lies in its incorporation of recent advances in diagnostic strategies, such as MRI-targeted biopsy and multivariable risk prediction models. Nonetheless, several limitations should be acknowledged. These include the possibility of language and publication bias, as the review was limited to studies published in English and Korean. Additionally, heterogeneity in study designs, population characteristics, and screening protocols may have affected the consistency of the findings. The generalizability of the results across different healthcare systems and cultural contexts is also limited. Despite these constraints, the review offers valuable insights into how public health nursing strategies can be leveraged to advance equitable, evidence-informed, and person-centered approaches to prostate cancer screening.

### 4.6. Future Research

Further research is warranted to refine prostate cancer screening strategies, particularly through longitudinal studies that assess long-term outcomes, cost-effectiveness, and patient-centered metrics such as quality-adjusted life years (QALYs). Future work should also explore the role of precision screening technologies including genetic risk profiling, artificial intelligence, and biomarker integration in diverse populations, especially among Asian cohorts [30]. In parallel, public health nursing interventions such as culturally tailored education, community-based outreach, and shared decision-making support should be evaluated for their impact on screening uptake and reduction in disparities. These interdisciplinary approaches are essential to ensuring that emerging screening strategies are not only clinically effective but also equitable and accessible.

## 5. Conclusions

From a public health nursing perspective, this review reinforces the critical role of CHWs in advancing equitable cancer prevention. CHWs are uniquely positioned to implement population-based screening education, culturally tailored counseling, and shared decision-making support particularly for high-risk groups and medically underserved men. In Korea, where prostate cancer incidence is rising but awareness remains uneven, CHW-led outreach and navigation programs can bridge access gaps and promote informed participation in screening. This systematic review synthesizes recent international and Korean evidence on the benefits, risks, and clinical applicability of prostate cancer screening tools. PSA-based screening demonstrates a modest but meaningful reduction in prostate-cancer-specific mortality, particularly among men aged 55 to 69. However, limitations—including overdiagnosis, overtreatment, and low specificity—highlight the urgent need for more individualized and accurate approaches to screening.

Emerging precision strategies, including multiparametric MRI, biomarker-based models, and AI-driven risk assessment, offer improved diagnostic performance and are increasingly supported by global guidelines. These findings underscore the importance of transitioning from universal PSA screening to age- and risk-stratified approaches that reflect both clinical utility and health system capacity. Furthermore, this review highlights the need to integrate culturally competent, community-based interventions—such as CHW-led and nurse-led education and shared decision-making support—to ensure that future screening programs are not only clinically effective but also equitable and person-centered. While adapting evidence-based screening guidelines to local contexts is essential, their effective implementation in Korea requires alignment with national infrastructure, demographics, and health policy systems to enhance outcomes and promote equity.

## Figures and Tables

**Figure 1 healthcare-13-01381-f001:**
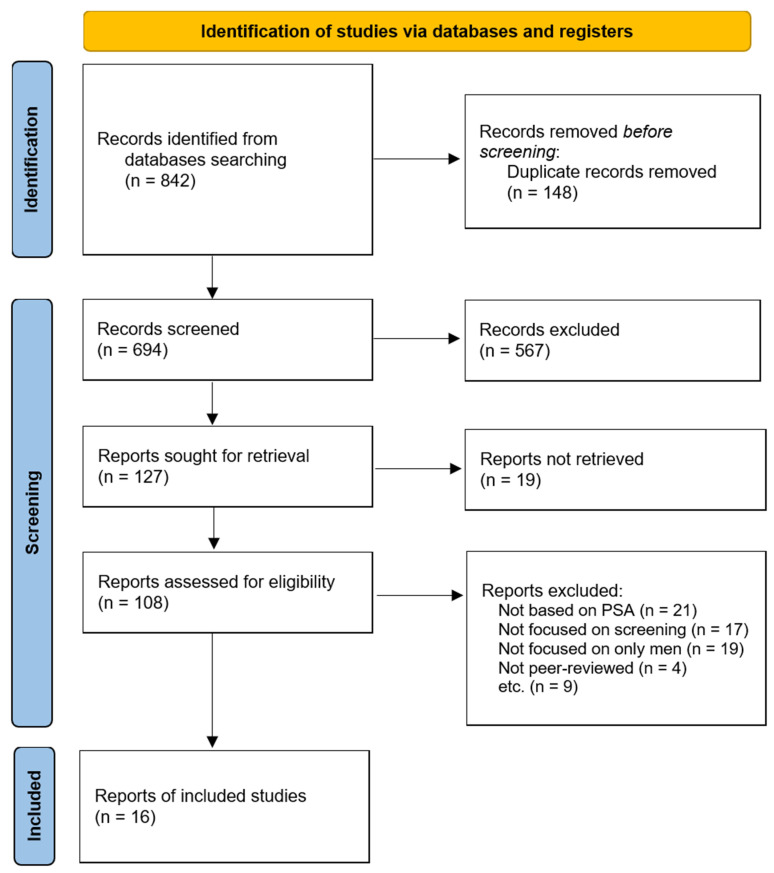
PRISMA flow diagram.

**Figure 2 healthcare-13-01381-f002:**
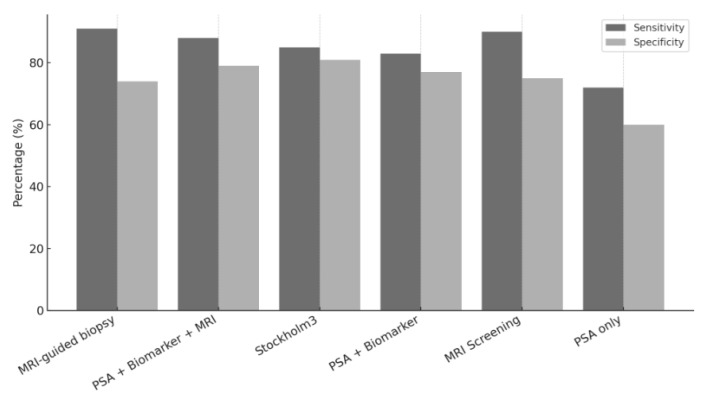
Comparison of diagnostic accuracy by method.

**Table 1 healthcare-13-01381-t001:** The characteristics of the studies included in the qualitative synthesis.

Reference	Year	Setting Country	Study Design	Sample	Key Findings
Auvinen A et al. [16]	2024	Finland	RCT	12,750	ProScreen trial: The evaluation of a multimodal screening protocol incorporating PSA, 4Kscore, and MRI demonstrated the improved detection of clinically significant prostate cancer.
Fazekas JT et al. [17]	2024	Multinational (primarily US)	Meta-analysis	80,114	MRI-based screening pathways maintained clinical significance in detecting relevant prostate cancer cases compared to conventional methods.
Björnebo et al. [18]	2024	Sweden	RCT	12,743	Biomarkers vs. MRI for prostate cancer screening: This study compared MRI-based and biomarker-based strategies for prostate cancer screening, highlighting their respective diagnostic efficacies.
Hao S et al. [19]	2022	Sweden	Simulation model	-	The Stockholm3 risk prediction model was found to be effective in the repeated screening and stratification of prostate cancer risk.
Klotz L et al. [20]	2021	Canada	RCT	453	MRI-targeted biopsy methods demonstrated superior sensitivity compared to traditional TRUS-guided biopsy methods.
Wei JT et al. [21]	2023	USA	Guideline	-	The AUA/SUO Early Detection Guidelines recommend personalized PSA-based screening strategies tailored to patient risk profiles and shared decision-making.
Bratt O et al. [22]	2023	Sweden, Europe	Narrative review	-	This review explored the long-term outcomes of prostate cancer screening and proposed future directions for policy refinement.
Harten MJ et al. [23]	2024	Europe	Narrative review	-	A comprehensive review of prostate cancer screening policies across Europe, focusing on variations in implementation and outcomes.
US Preventive Services Task Force [24]	2018	USA	Guideline	-	USPSTF PSA Screening Guidelines: PSA screening was shown to provide modest mortality benefits, though it was accompanied by risks of overdiagnosis and overtreatment.
Ilic D et al. [25]	2018	Europe, US, etc.	Systematic Review	721,718	The systematic review indicated both potential benefits in mortality reduction and harms related to overdiagnosis and false positives.
Martin RM et al. [26]	2018	UK	RCT	419,582	PSA Screening and 10-Year Mortality (CAP trial): A single invitation for PSA screening was associated with only a marginal reduction in long-term prostate cancer mortality.
Hugosson J et al. [27]	2019	Europe (8 countries)	RCT	182,160	ERSPC 16-Year Follow-Up Europe: The long-term data suggested a reduction in prostate-cancer-specific mortality following systematic PSA-based screening.
Martin RM et al. [28]	2024	UK, US, etc.	Meta-analysis	721,718	PSA screening reduced prostate-cancer-specific mortality but had no significant impact on overall mortality rates.
Vaccarella S et al. [29]	2024	Europe (26 countries)	Ecological study (registry data)	-	The study identified notable trends in the overdiagnosis of prostate cancer associated with widespread screening practices.
Mok Y et al. [30]	2015	South Korea	Cohort	97,274	Higher PSA levels at screening were associated with increased prostate cancer mortality in Korean men.
Pinsky P et al. [31]	2024	US	Cohort (registry data)	76,693	PSA screening reduced disease-specific mortality but was also linked to a high rate of overdiagnosis.

RCT = randomized controlled trial. PSA = prostate-specific antigen. TRUS = transrectal ultrasound.

**Table 2 healthcare-13-01381-t002:** Benefits and risks of prostate cancer screening tools.

Screening Tool	Benefits	Risks
PSA Test	Accessible, cost-effective	Overdiagnosis, possibility of false positive
PSA + MRI	High sensitivity, fewer biopsies	High cost, limited MRI access in some regions
Stockholm3/Biomarkers	Balanced accuracy, risk-stratified	Requires infrastructure, validation
TRUS biopsy	Widely used, standard method	Lower sensitivity, more complications

PSA = prostate-specific antigen. MRI: magnetic resonance imaging. TRUS = transrectal ultrasound.

**Table 3 healthcare-13-01381-t003:** Comparison of sensitivity and specificity by diagnostic method.

Study [Ref. No.]	Screening Tool	Sensitivity (%)	Specificity (%)
Klotz L et al. [20]	MRI-Guided Biopsy	91	74
Fazekas JT et al. [17]	MRI Screening	90	75
Auvinen A et al. [16]	PSA + Biomarker + MRI	88	79
Hao S et al. [19]	Stockholm3	85	81
Björnebo et al. [18]	PSA + Biomarker	83	77
Pinsky P et al. [31]	PSA Alone	72	60

MRI: magnetic resonance imaging. PSA = prostate-specific antigen.

**Table 4 healthcare-13-01381-t004:** GRADE summary of evidence certainty.

Outcome	Study Design	Risk of Bias	Inconsistency	Indirectness	Imprecision	Publication Bias	Overall Quality	Ref. No
Prostate-cancer- specific mortality	RCT/Meta-Analysis	Low	No serious	No serious	No serious	Unlikely	⬤⬤⬤⬤ High	[27,28]
All-cause mortality	RCT	Low	Serious	No serious	Serious	Unlikely	⬤⬤⬤◯ Moderate	[26]
Overdiagnosis rate	Meta-Analysis/Review	Moderate	Serious	No serious	No serious	Likely	⬤⬤◯◯ Low	[29]
Biopsy-related complications	Systematic Review	Low	No serious	Some	Serious	Unclear	⬤⬤◯◯ Low	[25]
Diagnostic accuracy (MRI/biomarkers)	RCT/Observational	Low	No serious	No serious	No serious	Unlikely	⬤⬤⬤⬤ High	[18,20]
Policy and guideline consistency	Guideline/Review	Low	No serious	No serious	No serious	Unlikely	⬤⬤⬤◯ Moderate	[21,23]

RCT = randomized controlled trial; ⬤ = criterion met; ◯ = criterion not met. Certainty levels: ⬤⬤⬤⬤ = High, ⬤⬤⬤◯ = Moderate, ⬤⬤◯◯ = Low, ⬤◯◯◯ = Very low (based on the GRADE approach).

## Data Availability

All data generated or analyzed in this review are publicly available via the Open Science Framework (OSF) at: https://osf.io/CR3YX (accessed on 1 April 2025). Additional information is available from the corresponding author upon reasonable request.

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
