# Peer review of "The Effectiveness and Harms of PSA-Based Prostate Cancer Screening: A Systematic Review"

_healthcare, 2025, doi:10.3390/healthcare13121381_

Round 1

Reviewer 1 Report

Comments and Suggestions for Authors

- This review paper lacks novelty, and covers scope that has been covered in other studies such as Cochrane systematic review (https://pubmed.ncbi.nlm.nih.gov/21392207/), https://www.bmj.com/content/362/bmj.k3519
- The focus on recent clinical trials is informative, but it does not progress the clinical understanding or provide any new insights
- This is a very narrative style of review with lack of statistical analysis on pooled data from different studies
- The CHW emphasis is not supported by original data and is more speculative
-Several sections are repetitive such as "Diagnostic Accuracy" vs "Effectiveness and Limitations", lots of overlaps in content. Figures and tables are not consistent with formatting.

Although the topic is important, this systematic review does not add anything new to the already available literature out there, review lacks methodological rigor, does not meet high-impact journal standards for systematic reviews, and therefore is not suitable for publication in its current form.

Author Response

Please review the attached file.

We believe that this review provides value by integrating recent clinical trial data, advances in diagnostic technology, and public health perspectives to offer practical insights for both clinical and policy researchers. We are committed to addressing the reviewers' concerns and improving the rigor and structure of the manuscript. If given the opportunity to revise, we will strive to ensure that the revised version meets the high standards of this journal for systematic reviews.

We appreciate your consideration of our manuscript for publication and your valuable feedback.

Reviewer 2 Report

Comments and Suggestions for Authors

The most important thing to consider during the editing process is the PRISMA guidelines. Although you state that your study followed the PRISMA guidelines, the PRISMA checklist and flowchart were not included in the manuscript. We recommend that you present these elements in detail throughout the paper as they are important for assessing the quality and transparency of a systematic review.

Several clarifications and improvements are needed to improve the clarity and completeness of the manuscript. In the abstract, it would be helpful to clarify that the observed mortality reductions apply specifically to prostate cancer-specific mortality, not overall mortality.

In the text, on line 75, where “new tools such as MRI” are mentioned, the claim would be more convincing if it were clarified whether MRI and related tools were an adjunct to or a substitute for PSA testing in the studies reporting these results.

Table 1 summarizes the included studies; it would be helpful to provide additional information such as “study design,” “country of origin,” “sample size,” and “outcomes evaluated in each study.

The citation format in the References section is inconsistent, and several entries are missing DOIs. Standardizing the format and including all relevant information would improve the professionalism of the manuscript.

Overall, your writing is clear, but you may benefit from professional English editing services to help maintain fluency and consistency throughout the text.

Author Response

We greatly appreciate Reviewer’s helpful input, which has significantly improved the clarity, transparency, and scholarly rigor of our manuscript. Once again, we sincerely thank Reviewer for their thoughtful and constructive feedback.

Reviewer 3 Report

Comments and Suggestions for Authors

Dear Authors,

The article requires major revisions before it can be considered for publication. Kindly see the points below for the details:

Abstract:

Line 11-12: Consider adding a data point to support the claim about increasing prostate cancer in Korea.

Line 14-15: Specify which six databases were searched to improve transparency.

Line 17-18: Clarify if the 20–31% reduction in mortality comes from one study or multiple sources.

Line 19: It’s unclear whether MRI and biomarker models were used in the reviewed studies or mentioned as future directions.

Line 21-22: Consider indicating whether the impact of education and shared decision-making was directly measured or inferred.

Introduction:

Line 30-31: The global incidence projection is sound, but I suggest briefly stating the source organization (e.g., GLOBOCAN) and citation for clarity.

Lines 32-33: Strong national context. However, consider clarifying whether the increase is due to improved detection, aging, or other factors.

Line 40-41: The 47.1% figure is impactful; please ensure the reference is up to date and methodologically sound.

Line 44: The statement about PSA being recommended needs clarification: “Is this a Korean national guideline or international consensus?”

Lines 47-49: The sentence introduces the controversy but could benefit from mentioning which major guidelines (e.g., USPSTF, EAU) differ.

Line 54-55: Consider specifying an example of an "advanced imaging modality" to enhance clarity for non-specialist readers.

Line 59-60: The public health nursing perspective is valuable. However, this shift in tone feels slightly abrupt; I recommend adding a sentence to transition from clinical to community relevance.

Line 27-66: The introduction section is too short and should be between 600 and 800 words.

Methods:

Line 76: Only five databases are listed, while six were mentioned in the abstract. This discrepancy should be corrected.

Line 84-85: It’s good that syntax adjustments are noted, but specifying examples for Embase or Cochrane would add clarity.

Line 89-90: Consider explicitly stating the population age range for consistency with other sections.

Line 164-165: The phrase [2 Guidelines/Advisory Statement, 2 Cohort study] needs grammatical correction for consistency (e.g., plural agreement).

Line 182: Acknowledge that blinding is inherently limited in PSA screening trials, but still explains how this could affect outcome interpretation.

Line 214-215: Please state how consistently these sensitivity/specificity values were reported across studies (single studies or aggregated data?).

Discussion:

Line 290-293: I would suggest that you clarify this review's unique contribution compared to prior ones, especially regarding the Korean context.

Line 295-301: Please mention whether the mortality benefit justifies screening from a cost-effective standpoint.

Lines 302-308: The harms are well-discussed. I recommend clarifying how frequently severe biopsy complications occur to contextualize their clinical weight.

Line 316-327: It might benefit from more specificity about how Korean guidelines currently differ from global ones.

Line 397-401: I recommend slightly rephrasing “adaptation of guidelines” to emphasize implementation feasibility in real-world Korean settings.

Additional points:

  • Methods and structure are appropriate, but minor tightening and clarification would benefit the narrative.
  • The emphasis on community health workers is valuable, though details of practical implementation could be expanded.

Author Response

We thank the reviewer again for their helpful insights, which have strengthened the clarity, specificity, and policy relevance of our manuscript.

Round 2

Reviewer 1 Report

Comments and Suggestions for Authors

The authors have extensively revised the manuscript and I really appreciate that. It needs minor grammatical edits, other than that the manuscript is good to go. 

Reviewer 3 Report

Comments and Suggestions for Authors

Dear Authors,

This manuscript has the potential for publication.